# High-Content Screening for the Detection of Drug-Induced Oxidative Stress in Liver Cells

**DOI:** 10.3390/antiox10010106

**Published:** 2021-01-13

**Authors:** María Teresa Donato, Laia Tolosa

**Affiliations:** 1Unidad de Hepatología Experimental, Instituto de Investigación Sanitaria La Fe, 46026 Valencia, Spain; 2Departamento de Bioquímica y Biología Molecular, Facultad de Medicina, Universidad de Valencia, 46010 Valencia, Spain

**Keywords:** oxidative stress, mitochondria, high-content screening, cell models, hepatotoxicity

## Abstract

Drug-induced liver injury (DILI) remains a major cause of drug development failure, post-marketing warnings and restriction of use. An improved understanding of the mechanisms underlying DILI is required for better drug design and development. Enhanced reactive oxygen species (ROS) levels may cause a wide spectrum of oxidative damage, which has been described as a major mechanism implicated in DILI. Several cell-based assays have been developed as in vitro tools for early safety risk assessments. Among them, high-content screening technology has been used for the identification of modes of action, the determination of the level of injury and the discovery of predictive biomarkers for the safety assessment of compounds. In this paper, we review the value of in vitro high-content screening studies and evaluate how to assess oxidative stress induced by drugs in hepatic cells, demonstrating the detection of pre-lethal mechanisms of DILI as a powerful tool in human toxicology.

## 1. Introduction

Drug-induced liver injury (DILI) is triggered by prescription and non-prescription drugs, as well as herbal products or dietary supplements, leading to liver impairment or damage, and in the worst case liver failure [1,2]. DILI is one of the most serious adverse drug reactions that challenges pharmaceutical companies, regulatory agencies and health care professionals. It is a relevant entity in medical practice, which carries significant morbidity and mortality, constitutes the major cause of hepatic dysfunction and acute liver failure in Western countries [3] and has been responsible for 32% of drug withdrawals between 1975 and 2007 [4].

Several hundreds of drugs have been associated with liver damage and the list is ever-growing [5]. For a few drugs (e.g., acetaminophen) liver injury is dose-dependent, predictable and experimentally reproducible. However, most DILI episodes are idiosyncratic, that is, unexpected adverse reactions occurring in a minority of patients at doses that are safe for the general population [6]. Although ideally drug toxicity should be discovered during preclinical testing, in practice, due to its rarity and unpredictability, hepatotoxicity is seldom detected in in vitro assays or animal models. Most DILI cases appear post-approval when the drug is administered to several thousand patients.

It has been reported that DILI can be a multistep and multicellular disease process with a wide range of chemical etiologies. Thus, a better understanding of the mechanisms underlying DILI is essential in order to evaluate current strategies for the detection of hepatotoxicity [7]. Several specific signaling pathways that are activated during DILI and that could be predictive for hepatotoxicity have been identified [8]. Among them, oxidative stress, which is the imbalance between the production of reactive oxygen species (ROS) and the ability of cells to confront them through antioxidant enzymes, has been described to play an important role.

During the past years, numerous in vitro systems have been described in the literature in an attempt to advance the understanding of the underlying mechanisms of human DILI [1,9]. These liver model systems include conventionally cultured hepatic cell lines, primary human hepatocytes (PHHs), co-cultures and engineered liver platforms such as bioprinted hepatic models or perfusion systems [10].

Additionally, readout technologies have also evolved from single mechanistic endpoints toward approaches that consider global changes produced by drugs. In this sense, toxicologists have begun to apply newer technologies including toxicogenomics, metabolomics and high-content screening (HCS) assays [1]. HCS assays are multiplexed cell staining tests developed to gain a better understanding of complex biological functions and mechanisms of damage to the liver or other tissues. They have become an important tool for the safety evaluation of drug candidates [11]. Moreover, HCS has been used not only for toxicity screening but also in order to elucidate the mode of action of drugs, which makes it a powerful tool in drug discovery and development.

In this paper, we review the use of in vitro HCS to assess oxidative stress induced by drugs in hepatic cells, demonstrating the detection of pre-lethal mechanisms of DILI as a powerful tool in human toxicology.

## 2. Mechanisms of Drug-Induced Hepatotoxicity

DILI is a complex multistep phenomenon which encompasses a wide spectrum of clinical presentations and may mimic any form of liver disease (acute and chronic hepatitis, steatohepatitis, phospholipidosis, cholestasis, cirrhosis, etc.) [5]. Mechanisms underlying drug-induced hepatotoxicity remain poorly understood, which represents a major obstacle for the reliable prediction of DILI cases, in particular the idiosyncratic ones. Advances in the mechanistic understanding of DILI reveal that drugs can induce liver damage through multiple mechanisms [12]. Thus, the application of mechanism-based integrated approaches contributes to more reliable predictions of potential hepatotoxic effects induced by drugs [13]. Impairment of mitochondrial function, induction of oxidative stress, depletion of the glutathione (GSH) pool, covalent binding to macromolecules, inhibition of apical hepatic transporters (i.e., the bile salt export pump) and activation by cytochrome P450 (CYP) enzymes have been proposed as mechanistic indicators of drug-induced hepatotoxicity [14,15].

The liver is recognized as a frequent target of toxic damage due to its anatomic location and its high content of drug metabolizing enzymes. Most orally administered drugs are metabolized in the liver before they reach other tissues. Although for many drugs metabolism renders non-toxic metabolites (detoxication), some drugs may undergo metabolic bioactivation, with the generation of reactive metabolites able to induce liver injury. Thus, the parent drug itself, or more often any of its metabolites, may directly induce a chemical insult to hepatocytes or may eventually initiate a series of processes (activation of immune response, release of inflammatory mediators, mitochondrial dysfunction, endoplasmic reticulum stress, etc.) which contribute to the progression of liver damage [6].

Drug metabolism reactions, in particular oxidations by CYP enzymes, play an important role in hepatotoxicity. Reactive metabolites generated during drug bioactivation may covalently bind to macromolecules to form adducts with DNA or proteins (Figure 1). Covalent modification of enzymes, transporters and other key proteins may result in metabolic dysfunction, altered signaling pathways, loss of cell homeostasis and cell death. Some modified proteins may also serve as potential haptens or new antigens that trigger immunologically mediated liver injury. Covalent binding is dependent on the proportion of the drug converted into a reactive metabolite, the half-life of the reactive intermediate and its ability to react with cell macromolecules [6].

Mitochondria play a pivotal role in the maintenance of energy metabolism, the integration of cell signaling pathways and the mechanisms of cell death. Thus, the alteration of mitochondrial homeostasis is a common cause of cell damage induced by drugs or their metabolites [16]. Inhibition of the mitochondrial respiratory chain (MRC), impairment of β-oxidation of fatty acids, induction of mitochondrial permeability transition pore opening and depletion of mitochondrial DNA are among the mechanisms involved in drug-induced mitochondrial dysfunction. Direct consequences of these disturbances of liver mitochondria are ATP depletion, lipid overaccumulation, the release of cytochrome c into the cytosol and increased ROS formation, which can result in steatosis/steatohepatitis, apoptosis or necrosis [17].

ROS are highly reactive molecules formed as a by-product of aerobic metabolism. They participate in diverse cell signaling pathways and modulate cell growth and metabolism [18]. Disturbances in redox homeostasis of the cell can result in increased ROS levels with deleterious effects on diverse cell components [19]. The mitochondrial respiratory chain (MRC) is the main cellular source of ROS and, under physiological and homeostatic conditions, low levels of ROS generated by electron transport activity can be easily neutralized by antioxidant defenses. However, rates of ROS production can increase in the mitochondria or in other cell compartments under some stressful conditions, such as drug-induced mitochondrial injury. An excess of ROS that are not adequately neutralized can lead to many harmful effects including oxidative damage to DNA, proteins and lipids [19].

## 3. Drug-Induced ROS Generation

Oxidative stress results from an imbalance between the generation of oxidant species and the antioxidant capacity of the cells. This situation is produced under different pathological conditions, including exposure to drugs that induce an excessive formation of ROS or a depletion of antioxidant defense systems (Table 1).

Although most electrons provided to the MRC are safely transferred to molecular oxygen to form water, direct or indirect impairment of the MRC by chemicals increase the occurrence of single-electron reductions of oxygen in the mitochondria to form ROS, mainly in the form of superoxide anions. The activity of superoxides dismutase and glutathione peroxidase, enzymes known to play important roles in the detoxification of ROS in the mitochondria, can be overtaken in some stressing circumstances. For instance, acetaminophen overdose leads to the formation of *N*-acetyl-p-benzoquinone imine (NAPQI), a reactive metabolite which depletes GSH, especially in mitochondria [19]. Thus, mitochondrial oxidative stress and peroxynitrite formation are induced, which trigger mitochondrial dysfunction, membrane permeability transition, mitochondrial depolarization, loss of calcium homeostasis and decreases in ATP synthesis [17]. An excess of oxidant species can cause a wide spectrum of oxidative damage to mitochondrial proteins, mitochondrial DNA and membrane lipids, which alters their structure and function. This oxidative damage aggravates mitochondrial dysfunction and further enhances ROS production, thus leading to important disturbances of cell bioenergetics and even cell death [17,19].

In addition to drug-induced disruption of the MRC, ROS can also be generated during metabolism of drugs by CYPs and other oxidative enzymes or by the presence of compounds that undergo repeated redox cycles. The formation of reactive intermediates by drug-metabolizing enzymes is involved in the hepatotoxicity of many compounds (Figure 1). Most reactive metabolites are electron-deficient species (electrophilic metabolites) that may react with nucleophilic sites in critical proteins or nucleic acids to form covalent adducts, leading to toxicity [6]. Alternatively, electrophilic metabolites form conjugates with GSH (via glutathione S-transferase activities) and deplete the GSH pool in the cell. There are many examples of CYP-mediated bioactivation of drugs into toxic metabolites in the endoplasmic reticulum of cells, such as the formation of electrophilic NAPQI from acetaminophen by CYP2E1, CYP1A2 and CYP3A4 [29]; the conversion of diclofenac into reactive quinone imines catalysed by CYP3A4 and CYP2C9 [12]; the formation of quinone and quinone methides from troglitazone by CYP3A4 [30]; and the oxidation of nitroaromatic moiety of flutamide by CYP1A2, CYP2C19 and CYP3A4 [26]. These bioactivation pathways are responsible for drug-induced hepatotoxicity via covalent binding and the formation of protein adducts, GSH depletion and oxidative stress [12].

Drug metabolism may also induce the formation of ROS or other radical species derived from oxidative or reductive metabolism of the drug. Oxidation of azathioprine (AZA) and other thiopurine drugs by xanthine oxidase has the potential to generate ROS that may result in liver damage via GSH depletion and mitochondrial injury [31]. Certain compounds increase intracellular levels of ROS through a cycling redox mechanism (Table 1). One-electron reduction of doxorubicin, an anthracycline anticancer drug, is carried out by several oxidoreductases to form a semiquinone radical that is re-oxidized back to doxorubicin with the formation of ROS [32]. Similarly, paraquat, a well-known toxicant to the liver and other tissues, undergoes one-electron reduction by diverse NADPH- or NADH-dependent oxidoreductases and is converted into a paraquat radical monocation. These radical species are then rapidly re-oxidized in the presence of molecular oxygen with the formation of a superoxide anion [33]. The result of these reactions is a futile redox cycle with the net regeneration of ROS leading to oxidative cell damage.

## 4. HCS Assays for the Detection of Oxidative Stress Induced by Drugs

### 4.1. HCS Technology

Multiparametric measurements of HCS assays are very useful in early toxicity and safety assessment in drug development. HCS technology, which allows automated image acquisition and analysis and provides information on multiple properties of individual cells loaded simultaneously with fluorescent dyes or transfected with fluorescent reporters, is used for drug safety evaluations. One of the main advantages of HCS technology is that distinct endpoints can be measured at a single-cell level simultaneously. A typical HCS assay for the detection of toxicity includes: (i) the selection of a relevant cell model; (ii) incubation with model test compounds; (iii) staining with a combination of fluorescent probes indicative of cell damage, and (iv) automated image acquisition and analysis using HCS equipment. Figure 2 summarizes the main steps in an HCS assay.

Different ex vivo and in vitro models have been used in HCS assays, from isolated primary hepatocytes to complex 3D models. Immortalized human cell lines such as HepG2 or HepaRG cells have been widely used for toxicological assessments as substitutes for PHHs in drug screening [13,34]. PHHs remain the gold standard in liver cell models, but they have many drawbacks like the scarcity of organs to isolate them and the loss of important enzymes and transporters when they are in culture [10]. To overcome these limitations, other models such as hepatocyte-like cells (HLCs) derived from induced-pluripotent stem cells (iPSCs) or more physiologically relevant 3D multicellular in vitro systems have been proposed [35]. An important technical issue in the first step of an HCS assay is to select an optimal cell density that allows an appropriate segmentation and the selection of plates that permit the best imaging.

The second step of an HCS assay for toxicology studies includes incubation with test compounds. In this phase, appropriate positive and negative controls may be properly selected to ensure that the assay allows the detection of the desired endpoint. These controls should be run in every set of experiments. Moreover, it should also be considered that some compounds can be autofluorescent and could interfere with some of the selected fluorescent probes.

HCS enables the multiplexing of the information that is obtained in a single assay, by using a combination of fluorescent probes. The major requirement for the use of multiple probes is their optical compatibility and the limitations of the instrument used for the measurements [36]. Section 4.2 describes in detail the most commonly used probes for oxidative stress detection in toxicological studies. Additionally, it should be considered that to get the brightest signal it is important to use illumination wavelengths that will optimally excite the fluorophore and to get the maximum of emission photons [37]. A comparison of the spectra of the fluorophores used and the fluorescent filter sets and/or laser line may be performed to ensure the optimal conditions [37]. It should be considered that some fluorescent dyes can be toxic to live cells, and therefore they can be only used with fixed cells.

After incubation with fluorescent probes, the automated acquisition of fluorescent images in separated channels is performed. Microscopy performance is usually assessed based on three criteria: sensitivity, resolution and speed [38]. It is impossible to have an optimal performance in each criterion, so a balance between them is pursued based on the compromise triangle of microscopy performance [38]. The selection of the objective affects the resolution, field of view and sensitivity, and should also be considered when setting up an HCS assay. Furthermore, considering that imaging a large quantity of cells to achieve high throughput is time-consuming [39], a correct focus is crucial for the correct development and use of HCS assays. Both image-based and laser-based autofocus methods are used in HCS. Image-based autofocus methods are slower, although they are better than laser autofocus for 3D structures [40]. Alternatively, machine learning methods are being developed to recognize and track only a subset of cells [40].

Due to the large numbers of images that are generated in HCS assays, automated image processing and analysis are decisive for large-scale image-based toxicity screenings. As a first step, in order to reduce noise and correct background or illumination, image-processing algorithms are used [37]. Secondly, cell segmentation is used to identify cellular or subcellular regions [41] and finally feature extraction allows the quantification of the changes in the identified regions [39], which in the case of toxicological assessment lead to the identification of significant changes produced by drugs compared to untreated cells. Thresholding is vulnerable to intensity variations from day to day and should be adjusted for day-to-day and user-to-user effects. On the other hand, accurate cell segmentation is a requisite to extract cell-by-cell information. Thus, the segmentation of touching cells is critical when analyzing HCS images [42]. Different algorithms, such as the evolving generalized Voronoi diagram [42], have been proposed in order to segment touching cells, which commonly have a variable morphology to improve the image analysis process.

Data analysis and management in HCS has advanced significantly in the past years, since the large volume of data produced in HCS assays is commonly a bottleneck in many projects. It should be considered that processing hundreds of images by applying image algorithms requires a powerful infrastructure [43]. Moreover, the data output of a 3D image-based screen is extremely high, which also poses a challenge in terms of data handling and image analysis [38].

### 4.2. HCS Probes for the Detection of Oxidative Stress

As ROS production is a major mechanism implicated in DILI, several fluorescent probes have been described for their use in toxicological studies. Fluorescent probes can simply detect ROS production in general or in specific radicals or locations. Table 2 summarizes the fluorescent probes and reporters for oxidative stress detection used in HCS.

For instance, the accumulation of the fluorescent compound 2′,7′-dichlorofluorescin (DCF), generated by intracellular oxidation of DHCF-DA, was used as an indicator of ROS generation, mainly H_2_O_2_ [53]. Once the membrane-permeant H_2_DCFDA enters a cell, its acetate moieties are cleaved by intracellular esterases, resulting in an impermeable H_2_DCF form. Subsequent oxidation of H_2_DCF produces the fluorescent 2′,7′-dichlorofluorescein, which can be detected in the green spectrum [53].

Monochlorobimane (mBCl) is a membrane-permeant probe that fluoresces in the UV spectrum upon reacting with GSH in a reaction catalyzed by the enzyme glutathione-S-transferase [54]. Due to the enzymatic catalysis of mBCl–GSH adduct formation, mBCl has greater specificity for GSH compared with other thiol-specific probes such as monobromobimane, which reacts freely with both GSH and intracellular protein thiols [54].

CellROX^®^ Oxidative Stress Reagents are fluorogenic probes designed to detect ROS in live cells. The CellROX Deep Red signal is localized in the cytoplasm of the cells and it has been used in several HCS assays for the detection of acute [45] or repeated-dose exposure to hepatotoxicants [51].

On the other hand, dihydroethidium (DHE), also called hydroethidine, dye is oxidized to fluorescent ethidium, which intercalates into DNA; the fluorescent signal is used as a measure of oxidative stress [49]. DHE exhibits blue fluorescence in the cytosol until it is oxidized, at which point it intercalates within the cell’s DNA, staining its nucleus a bright fluorescent red.

Lipid peroxidation, as a measure of oxidative stress damage, can be detected with the lipophilic probe, BODIPY 665/676 dye. This probe exhibits a change in fluorescence after interaction with peroxyl radicals [55].

Mitochondrial superoxide production can be identified by measuring MitoSOX Red fluorescent intensity in the mitochondrial compartment [50]. It has been used in different hepatic models for the study of the specific mechanisms implicated in DILI [50,51].

Other researchers have used HCS in reporter-cell lines. In this case, it is based on the use of a bacterial artificial chromosome containing a genomic copy of a particular gene and a fluorescent luciferase reporter construct introduced by homologous recombination [56]. Major cell stress signaling routes activated in response to toxicants include antioxidant response element activation; the heat shock response; the unfolded protein response; the metal stress response; the DNA damage response and the induction of phase I, II and III enzymes/transporters by nuclear receptors [56]. Cell-based reporters for the study of the oxidative stress induced by drugs Keap1, Nfr2 and Srxn1 have been used [57].

### 4.3. Examples of HCS Assays for the Detection of Oxidative Stress Induced by Drugs in Hepatic Cell Models

Multiple HCS assays have included fluorescent probes for the detection of oxidative stress damage induced by drugs, since oxidative stress is one of the major mechanisms implicated in DILI. For instance, we have demonstrated that HCS can be used for the detection of oxidative stress damage. Figure 3 exemplifies the utility of this technique using different fluorescent probes. AZA is a widely used immunosuppressive drug that is generally well tolerated but that may produce severe hepatitis in a small number of patients [58]. Oxidative stress and the subsequently activated immune- and inflammation-related factors have been suggested as major mechanisms in AZA-induced hepatotoxicity [59]. Here, we demonstrate that the oxidative response induced by AZA can be measured by HCS (Figure 3). Three different probes (MitoSOX Red, DCF and CellROX) were used and a dose-response was detected when HepG2 cells were incubated with a range of concentrations for 1 h. No significant effects were seen in viability, whereas a significant increase in ROS production was detected. DCF fluorescence seemed to be the most sensitive parameter, although significant dose-response changes were also detected with CellROX and MitoSOX Red. These representative results indicate the suitability of this technique to detect subtle changes even at subcytotoxic concentrations.

Many groups have used HCS for the detection of oxidative stress induced by drugs. Table 3 summarizes different studies that have used HCS for the detection of oxidative stress induced by drugs in different liver cell systems. In general, ROS production or GSH depletion are included in the HCS assays, as oxidative stress has been defined as a major mechanism implicated in DILI.

One of the first descriptions of the potential of HCS for the evaluation of DILI was the measurement of mitochondrial damage, oxidative stress and intracellular GSH levels in PHHs using TMRM, CM-H_2_DCFDA and mBCl, respectively, as fluorescent probes [48]. PHHs were incubated with 300 drugs at 100-fold of the therapeutic peak plasmatic concentration when possible. After 24 h incubation, they assessed the toxicity using HCS. The study showed a positive rate around 50–60% and a low false-positive rate (0–5%), which led many groups and companies to use it as a powerful tool in drug screening.

We have also included oxidative stress in our panels covering different mechanisms implicated in DILI [13,45]. We first described the use of HCS for developing and validating a cell-based protocol to assess the hepatotoxic potential of drugs and defining their mechanisms of action in HepG2 cells. In this assay, five different fluorescent probes to measure changes in mitochondrial membrane potential (TMRM), viability (PI), nuclear changes (Hoechst 33342), intracellular calcium concentration (Fluo-4 AM) and oxidative stress (BODIPY 665) were included. The assay turned out to be a simple screening that indicated the mechanisms implicated in both the toxicity and the degree of injury in a single incubation with high specificity and sensitivity [13]. Then, we applied the same strategy to a more complex test system, metabolically competent HepG2 cells, using CellROX as a measure of oxidative stress [45]. The assay allowed for the identification of bioactivable hepatotoxins and the mechanism(s) involved in their toxicity [45]. Finally, in order to assess the repeated-dose long term exposure to drugs, Upcyte human hepatocytes (UHH) were incubated with 15 toxic and non-toxic compounds for up to 21 days and the mechanisms implicated in their toxicity were assessed using HCS. In this case, three different probes for the study of oxidative stress-induced damage were used: mBCl, CellROX and MitoSOX Red [51].

Mennecozzi et al. (2015) investigated sex-specific differences in PHHs from different donors exposed to hepatotoxic drugs using HCS. They used a combination of probes which included DHE for ROS production. Differential effects in oxidative stress damage were detected depending on the compound assessed, although a general pattern could not be defined for female or male hepatocytes [49].

A comparative study of HepG2 and HepaRG cells for the prediction of DILI demonstrated that DILI compounds, which deplete GSH via reactive metabolites, showed a more significant decrease in GSH or increase in ROS in HepaRG than in HepG2 cells [62]. This study used two different panels for the detection of toxicity and included the oxidative stress probes such as CM-H_2_DCFDA for ROS production and mBCl for GSH depletion. Additionally, they pointed out differences in drug metabolism due to the significant differences between both cell models in GSH depletion, indicating the value of HCS assays for DILI screening and also for studying the contribution of metabolism to hepatocyte toxicity [62].

Since current in vitro test systems have different limitations for toxicological studies, other cell-based models, including stem cells differentiated into a hepatic phenotype, are being explored as a functional source of human hepatocytes. HLC-derived iPSCs have been used to evaluate the hepatotoxic potential of well-known drugs at subcytotoxic concentrations by means of HCS [61]. The authors used carboxy-H_2_DCFDA for the measurement of ROS and showed that ROS production was one of the most sensitive parameters for the detection of drug-induced toxicity, pointing out the suitability of this probe for the study of DILI in vitro [61].

On the other hand, the monitoring of adaptative stress response pathways has also been used as a predictive tool in toxicology [52,57]. Wink et al. (2016) established a panel of fluorescent protein reporter HepG2 cell lines using bacterial artificial chromosome (BAC) technology, reflecting three adaptative stress responses, and combined it with HCS technology. HCS allowed for the establishment of the temporal order of the activation of the distinct adaptative responses and the identification of the main ones induced by a selection of compounds that had previously been described to produce DILI [57]. The application of the HepG2 stress response reporters in 3D spheroid systems in combination with HCS has also been proposed [64]. HepG2 spheroids increase the expression of phase I and II metabolizing enzymes and show a high stability, which allows repeated-dose treatment. In this model, the authors demonstrated that a six-day repeated treatment with hepatotoxic drugs resulted in an increased sensitivity in the detection of cytotoxicity when compared to single-dose exposure, demonstrating their suitability for safety assessment [64].

Finally, HCS has been also used for the evaluation of the effects of natural products in hepatic cells, demonstrating the suitability of this tool for assessing both oxidative stress induction [65] and the antioxidant capacity of compounds [66].

HCS may be used in the pharmaceutical industry to screen candidate drugs for potential human toxicity, to rank and select lead compounds and to demonstrate the major mechanisms of toxicity [67]. However, despite their great sensitivity and specificity in identifying human DILI, HCS assays fail to provide translational biomarkers. Thus, HCS multiparametric assays could be used in combination with other highly sensitive approaches, such as expression analyses [68] or toxicometabonomics [69], that could contribute as complementary assays to better predictivity and sensitivity of the human hepatotoxicity potential for new compounds.

## 5. Conclusions and Future Outlook

The high financial investment required to bring a drug to the market, along with the high rates of attrition of new molecules, make necessary the optimization of screening procedures for more efficient evaluations of drug safety. Human cell models are increasingly used in preclinical safety testing, as they provide quick and relatively inexpensive information in the early phases of drug development. As previously stated, the mechanisms responsible for DILI are poorly understood, which complicate the establishment and validation of predictive screening tools and limit the reliability of in vitro toxicity testing. Traditional tests rely on the determination of unspecific endpoints indicative of overt loss of cell integrity and have been shown to be insufficient to predict the hepatotoxicity of many drugs. Multiparametric HCS or omics-based approaches have gained popularity for hepatotoxicity studies as they enable the simultaneous determination of multiple informative parameters that may contribute to more accurate DILI predictions. In particular, HCS allows for the simultaneous analysis of several markers of cell damage and enables the detection of subtle toxicity-related changes with greater sensitivity than conventional single-endpoint cytotoxicity assays.

The flexibility of HCS technology allows researchers to easily adapt assays to multiple applications by selecting the most appropriate cell model and fluorescent probes. HCS has been proven as a valuable screening tool for the identification of hepatotoxic drugs in diverse cell models (PHHs, hepatoma cell lines, UHHs, adenoviral-transfected cells, HLCs). Cell-based HCS assays have been evaluated and validated in the toxicological field not only for the study of hepatotoxicity but also for neurotoxicity, cardiotoxicity and nephrotoxicity applications [70]. A careful selection of the most appropriated probes allows the optimization of specific assay panels covering a wide spectrum of mechanistic parameters. In this line, many fluorescent labeling probes are available for oxidative stress detection and have been successfully applied to HCS assays in several cell systems. Oxidative stress is one of the main mechanisms implicated in the toxicity of many drugs, and ROS generation and GSH depletion are considered among the most sensitive parameters of drug-induced hepatotoxicity [61,62]. HCS-based testing allows multiplexed measurements in cells cultured in multi-well plate formats. Thus, there are considerable reductions in time, cells, reagents and tested compounds needed for hepatotoxicity evaluations, rendering the assays amenable to medium/high-throughput screenings in early drug development steps.

One potential limitation of traditional monolayer cell systems used in hepatotoxicity evaluations is the loss of the native 3D microenvironment of liver cells. Recent technological advances have allowed the development of more complex in vitro models that better reproduce the spatial organization and inter-cellular interactions in the liver [71]. 3D culture systems stabilize many hepatocyte functions and seem to be a valuable alternative for long-term toxicity studies. The application of readout technologies to cellular models that allow repeated exposure assessments to mimic the prolonged clinical exposure in patients, and not only acute liver damage, is a promising strategy to improve DILI predictions. Thus, the combination of mechanistic multiparametric HCS measurements and physiologically relevant culture models will provide potent screening platforms for human toxicity.

The widespread application of HCS assays to the study of oxidative stress induced by drugs could provide fast, sensitive and accurate safety evaluations in early drug development. Cell-based HCS tools could also enhance the understanding of mechanisms and pathways involved in oxidative cell damage and could contribute to the identification of drugs, natural products or foods with antioxidant capacity.

## Figures and Tables

**Figure 1 antioxidants-10-00106-f001:**
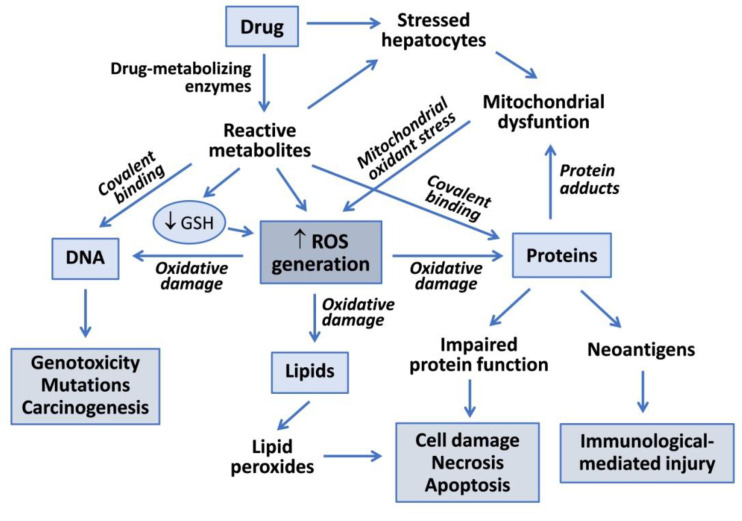
Mechanisms of drug-induced hepatotoxicity. Drug metabolism can render reactive metabolites that are able to induce hepatotoxicity via covalent binding to macromolecules or via oxidative damage induced by increased reactive oxygen species (ROS) generation. Reactive metabolites can interact with DNA to produce genotoxicity, induce lipid peroxidation or form adducts with proteins, which can lead to functional impairment, cell death or to the generation of neoantigens that trigger immune-based toxicity. Mitochondrial dysfunction induced by drugs, or more frequently by their reactive metabolites, contributes to increased ROS formation and oxidative damage to liver cells.

**Figure 2 antioxidants-10-00106-f002:**
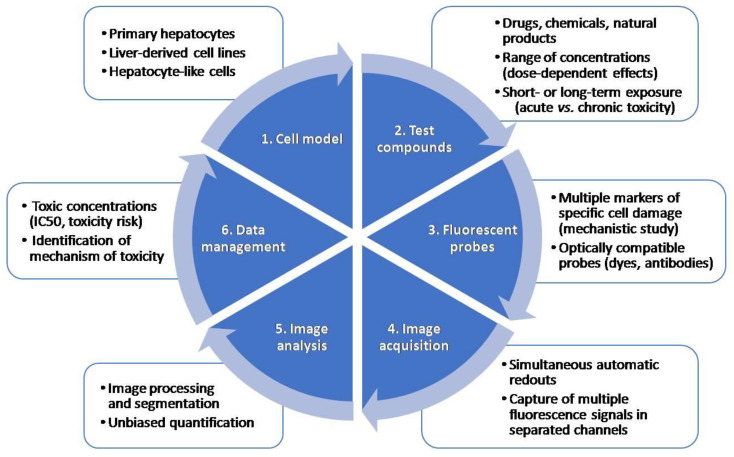
Major steps in a high-content screening (HCS) assay. An HCS assay includes the selection of an appropriate test system, incubation with test compounds at a range of concentrations, incubation with a combination of fluorescent probes, automated image acquisition, analysis and data management.

**Figure 3 antioxidants-10-00106-f003:**
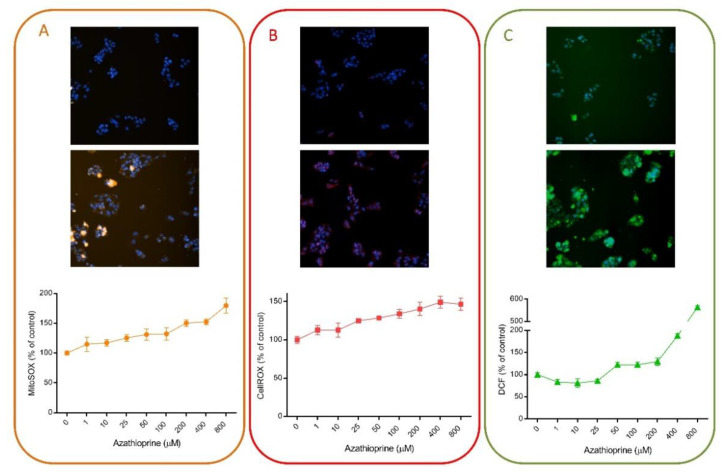
Azathioprine induces oxidative stress in liver cells. HepG2 cells were exposed to a range of concentrations of azathioprine (1–800 µM) for 1 h and then, incubated with MitoSOX Red (**A**), CellROX Deep Red (**B**) or DCF (**C**) and were analyzed using HCS. On the top of each panel, non-treated cells are shown, in contrast to cells treated with 400 µM of azathioprine, which are shown in the bottom part. HCS technology allows the quantification of changes induced by drugs and shows its utility for toxicological studies and to determine the major mechanisms implicated in their toxicity.

**Table 1 antioxidants-10-00106-t001:** Examples of toxic compounds able to induce oxidative stress and mechanisms involved.

Mechanism	Compounds	References
Direct or indirect impairment of the mitochondrial respiratory chain (MRC)	Acetylsalicylic, amiodarone, azathioprine, buprenorphine, chloroquine, lovastatin, tamoxifen, nefazodone, troglitazone	[12,16,20,21]
Depletion of GSH pool and/or antioxidant enzymes	Azathioprine, doxorubicin, flutamide, isoniazid, valproic acid, paracetamol (acetaminophen)	[16,19,22,23,24,25]
Generation of electrophilic metabolites	Amitriptyline, benoxaprofen, diclofenac flutamide, paracetamol, ticlopidine, trogitazone	[12,13,16,25,26]
Redox cycling-induction	Diquat, paraquat, menadione, doxorubicin, flutamide	[22,27,28]

**Table 2 antioxidants-10-00106-t002:** Probes and reporters used in HCS assays for the detection of oxidative stress.

Probe/Reporter	Indicator	Excitation	Emission	Reference
BODIPY 665/676	Peroxyl radicals (lipid peroxidation)	665	676	[13]
CellROX Green	ROS production (nuclei)	485	520	[44]
CellROX Deep Red	ROS production (cytoplasm)	644	665	[45]
CM-H_2_DCFDA		495	527	[46,47,48]
Dihydroethidium	Superoxide anion	518	605	[49]
mBCl	Glutathione	390	478	[46,47,48]
MitoSOX Red	Mitochondrial superoxide	510	580	[50,51]
Srnx1-GFP reporter	Nrf2 oxidative stress response	488	510	[52]
CHOP-GFP reporter	Endoplasmic reticulum-stress/unfolded protein response	488	510	[52]
p21-GFP reporter	P53 dependent DNA damage-related signalling	488	510	[52]
ICAM1-GFP reporter	NF-κB-mediated pro-inflammatory cytokine signalling	488	510	[52]

**Table 3 antioxidants-10-00106-t003:** HCS assays for the detection of oxidative stress-induced liver injury. DILI: drug-induced liver injury; HLCs: hepatocyte-like cells; iPSCs: induced pluripotent stem cells; PHH: primary human hepatocytes; UHH: Upcyte human hepatocytes.

Test Model	Probes/Reporters	Drugs	Reference
PHH	DRAQ5; TMRM; CM-H_2_DCFDA; mBCl	300 DILI and non-DILI compounds	[48]
HepG2	Hoechst 33342; PI; TMRM; Fluo-4 AM; BODIPY 665/676	78 DILI and non-DILI compounds	[13]
HepG2 transfected with CYP adenovirus	Hoechst 33342; PI; TMRM; Fluo-4 AM; CellROX	15 DILI and non-DILI compounds	[45]
HepG2, HepG2 + S9, PHH	CM-H2DFFDA, TMRE Activated caspase-3, phosphorylated histone- H3 and HSP 70/72 assays LipidTox (Phospholipids + neutral lipids)	144 DILI and non-DILI compounds	[60]
Male and female PHHs	DHE, TMRE, TOTO3, Fluo4	6 chemicals	[49]
iPSC-HLCs	MitoTracker orange, carboxy-H2DCFDA, TOTO-3, Hoechst33342	8 toxicants	[61]
HepG2 and HepaRG cells	Hoechst 33342, CM-H2DCFDA, TMRM, TOTO-3 DRAQ5, mBCl, YOYO-1	28 DILI and non-DILI compounds	[62]
HepG2	Cellomics HCS reagent; CellROX Deep Red; Hoechst 33342	Complex mixtures of perfluorinated, brominated, and chlorinated compounds (persistent organic pollutants)	[63]
HepG2 reporter lines	11 BAC-GFP reporters containing target genes, representing the oxidative stress response pathway (KEAP1/NFR2/SRXN1), the unfolded protein response and DNA damage	30 hepatotoxicants	[57]
HepG2 reporter lines	SRXN1-GFP, CHOP-GFP, p21-GFP and ICAM-GFP	118 FDA-labelled drugs	[52]
HepG2 reporter lines in 3D spheroids	Srxn1-GFP, NQO1-GFP, BiP-GFP, Chop-GFP, p21-GFP and Btg2-GFP	33 compounds	[64]
UHH	BODIPY 493/503; LipidTOX Red Phspholipidosis, CellROX, Fluo-4 AM, Hoechst 33342; PI; MitoSOX Red, TMRM and mBCl	15 DILI and non-DILI compounds	[51]

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
