# Peer review of "High-Content Screening for the Detection of Drug-Induced Oxidative Stress in Liver Cells"

_antioxidants, 2021, doi:10.3390/antiox10010106_

Round 1

Reviewer 1 Report

Generally the manuscript is well written.

-the quality of figure 3 should be improved.

-authors should also discuss the limitations of these techniques for completeness of information.

-the authors could also better discuss the procedural and technical aspects of these assays

Author Response

We thank the reviewer for the constructive nature of their comments and suggestions on the manuscript, which have helped us to improve our resubmission.

-The quality of figure 3 should be improved.

Figure 3 has been also uploaded as a 600 dpi image as requested in the instructions of the Journal.

-Authors should also discuss the limitations of these techniques for completeness of information.

Following reviewer’s suggestion, we have included the limitations of HCS (lines 402-409) as well as new references (67-69) to support this point.

-The authors could also better discuss the procedural and technical aspects of these assays

Different important technical aspects to these assays have been now included for a better understanding. These critical aspects have been included in the description of each step of a HCA assay (lines 207-215; lines 224-226; lines 231-233; lines 246-247) to address reviewer’s comment.

Reviewer 2 Report

In my opinion the paper is quite interesting and merits publication in Antioxidants.

Minor revisions:

In some sentences English needs to be revised to reach the language standards that are required by the journal.

Furthermore, all typing errors should be corrected.

Author Response

We thank the reviewer for the constructive nature of their comments and suggestions on the manuscript, which have helped us to improve our resubmission.

  • In some sentences English needs to be revised to reach the language standards that are required by the journal. Furthermore, all typing errors should be corrected.

Grammatical errors have been corrected and English language edited.